# Healthy Eating Enhances Intrinsic Capacity, Thus Promoting Functional Ability of Retirement Home Residents in Northern Taiwan

**DOI:** 10.3390/nu14112225

**Published:** 2022-05-26

**Authors:** Kian-Yuan Lim, Hui-Chen Lo, In-Fai Cheong, Yi-Yen Wang, Zi-Rong Jian, I-Chen Chen, Yun-Chun Chan, Shyh-Dye Lee, Chi-Chun Chou, Feili Lo Yang

**Affiliations:** 1Ph.D. Program in Nutrition and Food Science, College of Human Ecology, Fu Jen Catholic University, New Taipei City 242062, Taiwan; limkianyuan@gmail.com; 2Department of Nutritional Science, College of Human Ecology, Fu Jen Catholic University, New Taipei City 242062, Taiwan; 041663@mail.fju.edu.tw (H.-C.L.); infaiinfai@gmail.com (I.-F.C.); bebeeateat1005@gmail.com (Y.-Y.W.); qmio137@gmail.com (Z.-R.J.); believe850617@gmail.com (I.-C.C.); jenny850823@gmail.com (Y.-C.C.); 3Department of Family Medicine, Fu Jen Catholic University Hospital, New Taipei City 243089, Taiwan; shyhdye@gmail.com; 4Department of Otorhinolaryngology, Yonghe Cardinal Tien Hospital, New Taipei City 234408, Taiwan; cthyh10680@gmail.com

**Keywords:** healthy aging, plant-based dietary supplements (PDS), Taiwanese Healthy Eating Index (T-HEI), texture-modification, vitality

## Abstract

Healthy aging is defined as the process of developing and maintaining functional ability in older age with intrinsic capacity, the composite of all the physical and mental capacities of an individual, being the core. This study was conducted to explore the intervention effects of improved dietary quality on intrinsic capacity. A prospective single-group interventional quasi-experimental study with 59 functional independent older adults from retirement homes were recruited. Texture-modified plant-based dietary supplements were provided. In addition, dietary intake, functional ability, and intrinsic capacity in vitality, locomotion, cognition, and psychological capacity were assessed. Vitality was captured by nutritional status, muscle strength, and cardiorespiratory endurance. Locomotor capacity was assessed based on the performance of physical fitness in backscratch test, chair-sit-and-reach test, chair-stand test, one-foot-standing test, and gaits peed. Psychomotor capacity and cognition were measured by using 15-item Geriatric Depression Scale (GDS-15) and Mini-Mental State Examination (MMSE), respectively. In a 4-month of intervention, after controlling for baseline values and covariates, participants with higher dietary intervention adherence showed a significant improvement over time in vitality captured by cardiorespiratory endurance (*P*_interaction_ = 0.009) and significant improvement in locomotion captured by gait speed (*P*_clusters_ = 0.034). A significant decrease in the chair-stand test (*P*_time_ = <0.001) and MMSE (*P*_time_ = 0.022) was observed during the four months of intervention. Enhanced intrinsic capacity further contributed to the improvement of ADL over time (*P*_interaction_ = 0.034). In conclusion, healthy eating enhances intrinsic capacity in vitality and locomotion thus promoting functional ability among older adults.

## 1. Introduction

People are living longer these days with the average global life expectancy surpassing 73 years in 2019 [1]. In Taiwan, the average life expectancy was 81.3 years in 2020. The share of the older population (≥65 years) is going through steady but rapid growth and will likely swell from 3.3 million (14% of the entire population) in 2018 to 5 million (≥20% of the entire population) by 2025 with a dramatic increase in the very old population aged 80 and above [2,3]. The increase in longevity is mainly due to declining mortality among older adults. Yet, there is uncertainty about whether people stay healthy or suffer poor health during those additional years lived [1,4]. Growing old is associated with a wide variety of underlying physical and psychological changes that result in a decrease in functional reserves. The gradual decrease in reserves leads to increased susceptibility to diseases and impairment. However, health in older age is complex. The distinction between “health” and “disease” has become increasingly ambiguous when living with well-controlled multimorbidity that has little impact on functional ability becomes a norm with the advancement of healthcare and medical technology. However, functional ability loss caused by physical, visual, and cognitive impairments contributes to the biggest causes of the burden of disease among older adults [4,5].

Shifting from the traditional disease-oriented focus to an innovative functional perspective approach, healthy aging was defined as the process of developing and maintaining intrinsic capacity that enables well-being in older age by the World Health Organization (WHO) in 2015 [6]. Opposite to frailty which represents deficits accumulated with aging, intrinsic capacity is a multidimensional indicator that combines the functional reserves in five different domains related to healthy aging including the vitality, locomotion, cognition, psychological capacity, and sensory capacity of an individual. The interaction between intrinsic capacity and relevant environmental characteristics determines the functional ability of an individual, which is the key to healthy aging [7]. Emerging evidence suggests that domains of intrinsic capacity are interrelated, and the decline of intrinsic capacity in older age was associated with an increased risk of adverse health outcomes such as frailty, disability, nursing home stay, and even mortality [8,9,10,11,12,13]. Previous studies on intrinsic capacity have focused on the trajectory of functioning. Studies on capacity-enhancing approaches are scarce.

Aging is characterized as a progressive accumulation of molecular and cellular damage which is associated with the deleterious change in the oxidative defense system and immune function. Unregulated oxidative stress and inflammaging further result in an increased risk of diseases and trigger a decrease in functional reserves [14,15]. Nutrition is a vital cornerstone in healthy aging [16]. Fulfilling energy and nutritional needs through diets of high quality is central to maintaining the function of the antioxidative defense system and immunity, as well as crucial in disease and impairment prevention. Adhering to a healthy diet is observed to be a contributing factor to greater intrinsic capacity [17,18,19,20,21]. Improving diet quality through dietary intervention to foster intrinsic capacity, however, has not been investigated. This study, therefore, aimed to explore the effect of improved diet quality, a capacity-enhancing approach, on vitality, locomotion, cognition, and psychological intrinsic capacity.

## 2. Materials and Methods

### 2.1. Data Acquisition

The data analyzed in this study were derived from the integrated research titled “New Strategies for Developing Healthy, Aging and Happy Diet”, which was funded by the Ministry of Science and Technology (MOST), Taiwan. Consisting of three subprojects, Subproject 1 of the integrated research focused on the development of texture modification food products. Subproject 2 was responsible for the human study, which rendered the data used in this study. Lastly, the biochemical assessment was conducted in Subproject 3 of the collaboration (Figure 1).

### 2.2. Sampling and Study Design

A quasi-experimental study with a prospective single-group intervention design was conducted between September 2018 and January 2019 in two retirement homes in northern Taiwan. Orientations regarding the study procedure, assessment, and intervention were held prior to the study in both retirement homes. An initial sample of sixty-five older adults aged 65 and above from the retirement homes were invited to take part in this study. After excluding participants who were visually and functionally disabled (*n* = 2), a caregiver for the resident who did not live in the retirement home (*n* = 1), a candidate who relocated before the start of the study (*n* = 1), and those who declined participation in the study (*n* = 2), a total of 59 cases provided with written informed consent were recruited. Approval of the study protocol was granted by the Institutional Review Board of Fu Jen Catholic University (IRB certificate number: C106019).

### 2.3. Dietary Intervention and Compliance

Four different plant-based dietary supplements (PDS) rich in vitamins such as vitamin A, vitamin B1, vitamin C, and minerals including potassium, calcium, magnesium, and phytonutrients were provided to the participants during the 4-month intervention period. Each serving of the PDS contained 1.5 exchanges of nuts and seeds and 1 exchange of two different fruits and vegetables, respectively. Since fruits and vegetables such as apple, pear, carrot, and Chinese kale, as well as nuts and seeds used in PDS, are firm while edentulism is common among older adults. Besides, the development of texture modification food products was one of the aims of the integrated project. Therefore, the PDS was texture-modified using a micronization process by a contracted manufacturer into liquidized and pureed forms in conformance with the International Dysphagia Diet Standardisation Initiative (IDDSI) Grade 3 and Grade 4 standards [22]. The texture-modified PDS could be served as smoothies or soup despite the participants in our study were not suffered from dysphagia. Detail on the nutritional label of each PDS was provided in Appendix A and Table A1, and has been reported elsewhere [23]. In brief, all participants were contacted personally and provided with 4 servings of different PDS weekly. Participants were instructed to consume all the PDS in addition to their usual intake. Consumption details were recorded and unconsumed PDS from the previous week were retrieved and recorded by research assistants. Additionally, weekly educational workshops on nutritional health, specifically discussing daily food guides (DFGs) and dietary guidelines (DGs) recommendations, were conducted in order to promote study compliance and to encourage behavioral change among the participants.

In addition to PDS provision, meal plan modification was carried out in collaboration with a dietitian in the retirement home so that the suggestion of DFGs and DGs were adhered to thus, improving the quality of the diet. For example, one-third of the vegetables were replaced with nutrient-dense dark-colored vegetables. Under a reasonable budget limit, mushrooms rich in vitamin D were also suggested.

### 2.4. Intrinsic Capacity Assessment

Intrinsic capacities in vitality, locomotion, cognition, and psychological capacity were determined at baseline, 2 months, and 4 months of intervention, respectively. Vitality was captured by nutritional status assessed by Mini Nutritional Assessment (MNA), a handgrip strength (HGS) test, and cardiorespiratory endurance assessed through a two-minute step-in-place test (TMS). Locomotor capacity was assessed based on the performance of physical fitness in a backscratch test (BST), chair-sit-and-reach test (CSR), chair stand test (CST), one-foot-standing test (OFS), and a three-meter timed-up-and-go test (TUG). Psychological capacity and cognition were assessed by using the 15-item Geriatric Depression Scale (GDS-15), and the Taiwan version Mini-Mental State Examination (MMSE) [24,25,26,27].

The physical fitness assessment was administered by licensed physical fitness trainers (K.-Y.L. and Z.-R.J.) and exercise professionals with the American College of Sports Medicine (ACSM) Exercise is Medicine^®^ (EIM) Taiwan credential (K.-Y.L., Y.-Y.W., Z.-R.J., I.-C.C. and Y.-C.C.) according to the Regulations of Physical Fitness Assessment for accurateness assurance [25]. Physical fitness trainers and exercise professionals were asked to carefully monitor the participants to ensure their safety throughout the assessment session. Assessments such as MNA, GDS-15, and MMSE were carried out by licensed dietitians (Y.-Y.W. and Y.-C.C.) and a certified dementia case manager (K.-Y.L.).

The nutritional status was evaluated by using the Mini Nutritional Assessment (MNA), a test composed of brief questions and measures on anthropometric measurements, including weight, height, and weight loss; global assessment related to lifestyle, drug prescription, and mobility; dietary assessment regarding the number of meals, fluid, and food intake, and feeding autonomy; and a subjective assessment on self-perceived health and nutritional status. A score greater than 24 suggests a desired nutritional status; 17 to 23.5 suggests a risk of malnutrition; a score below 17 suggests malnutrition [24].

HGS of the dominant hand was measured by using a JAMAR^®^ hydraulic hand dynamometer in a seated position with the best result of three attempts recorded to the nearest 0.1 kg.

TMS measured the participants’ cardiorespiratory endurance. Before the assessment began, a mark corresponding to the midway between the patella and iliac crest was placed next to the participants. Participants were instructed to march in place with their knees raised to the height of the mark as many times as possible for two minutes. Rest was allowed during the test, but the time would keep running until the conclusion of the test at two minutes. Counts of the right knee reaching the required height were recorded [25].

The BST measured flexibility in the shoulder joint. Participants started the assessment in a standing position with one hand placed behind the back, palm facing outward and fingers pointing upward, moving up along the spine toward the head. The opposite hand reaches up over the shoulder with the palm facing inward and fingers directed downwards. The procedure was repeated for the other hand and the best result was recorded. The distance in centimeters between the tips of the middle fingers was recorded as zero if the fingertips touched, positive/+ for overlapped, and negative/− for no touching, to the nearest 0.5 cm [25].

The CSR measured the flexibility of the lower back and hamstring. Participants sat on the edge of a chair with a 43-cm seat height with one-foot flat touching the floor and the other leg extended forward, knee straight, heel on the floor, and toes directed upward; one hand was placed on the top of the other hand with the middle fingertips even. Participants were instructed to reach forward toward the toes as much as possible in a slow and steady movement without bouncing and with the knee of the extended leg kept straight, holding for 2 s. Participants were allowed a try each leg to decide which leg to extend. The distance in centimeters between the tip of the middle fingers and toes was recorded as zero if the fingertips touched, positive/+ for overlapped, and negative/− for no touching, to the nearest 0.5 cm [25].

CST was used to measure lower extremity strength. Participants, sitting on a stabilized chair with a 43-cm seat height, were instructed to stand up from a seated position with arms folded across the chest. The number of full stands completed within a 30-s interval was recorded [25].

For the three-meter TUG test, participants started the test in a seated position, hands on the knees and feet flat on the floor on a chair with a 43-cm seat height with a cone placed 3 m in front of the chair. Participants were asked to get up and walk, not run, as fast as possible to go around the cone and return to the seated position. Two trials were performed, and the best time it took to complete the test was recorded to the nearest 0.1 s. Gait speed (meters per second, m/s) was calculated as distance completed (i.e., 6 m) divided by time spent [25].

In the OFS test, participants were instructed to stand unassisted on one leg with arms akimbo with the big toe of the free leg placed against the ankle of the standing leg. Participants were allowed to test each leg once to choose the standing leg. The duration the participants spent in the initial balanced position was recorded to the nearest 0.1 s with a maximum of 30 s. Times exceeding 30 s were recorded as 30 s [25].

Self-reports of depression were assessed using the Geriatric Depression Scale-short form (GDS-15), a 15-item instrument that screens for depressive symptoms among the elderly. A score greater than 5 suggests depression [26].

MMSE, an assessment tool with 30 items comprising subscales assessing the abilities in orientation, word registration, attention, word recall, language, and visuospatial was used to assess general cognitive performance. A score greater than 24 suggests normal cognitive performance [27].

### 2.5. Dietary Assessment and Dietary Quality Appraisal

Five random single-day dietary records were collected monthly during the intervention period. Additional dietary choices/preferences regarding the meal provided in the retirement homes were collected. The frequency of whole-grain choices and meals away from retirement homes were investigated; detailed information on the general intake of staple food, main dishes, side dishes, and vegetables, whether normal serving, half serving, or double servings, was also documented.

Diet quality was assessed monthly throughout the intervention period using the Taiwanese Healthy Eating Index (T-HEI), a novel healthy eating appraisal system that examines the dietary adherence to Taiwanese DFGs and DGs, based on dietary records collected. Detail on the T-HEI was described elsewhere [28]. Briefly, T-HEI is a 100-point scale measuring compliance to the dietary guidance of Taiwan, which includes adequacy and moderation components. For ensuring nutrient sufficiency, intake of whole fruits (10 points), total vegetables (5 points), dark or orange vegetables (5 points), whole grains (10 points), total protein foods (5 points), plant proteins and seafood (5 points), dairy (10 points), fatty acids (5 points), as well as nuts and seeds (5 points) at the recommended level or above was given a maximum score, while no intake was given a minimum score of zero. Intake between none and the maximum standard was scored proportionally. The reverse scoring method was used for the consumption of saturated fats (10 points), refined grains (10 points), sodium (10 points), and alcohol (10 points) for chronic disease prevention purposes. T-HEI was calculated for meals provided in the retirement homes (T-HEI_Diet_) and for meals provided in the retirement homes with the addition of dietary supplements consumed (T-HEI_Diet+PDS_).

### 2.6. Outcome Assessment

The functional ability on the activity of daily living (ADL) and the instrumental activity of daily living (IADL) were assessed as the interventional outcomes. The Barthel Index was used to assess the participants’ capability in self-care activities including self-feeding, bathing and showering, personal hygiene and grooming, dressing, bowel and bladder control, toilet use, as well as functional mobility [29]. The Lowton & Brody Index was used to assess the participants’ abilities in independent living, including telephone use, shopping, food preparation, housekeeping, laundry, transportation, taking prescriptions, and finances management [30].

### 2.7. Statistical Analysis

A hierarchical cluster analysis with Ward’s linkage method was used to group participants according to their dietary adherence similarity. T-HEI_Diet+PDS_ scores across the study period were standardized using Z-score for clustering variables. Three clusters were determined through visual inspection. The mean and standard error of the mean (SEM) were reported for continuous variables. The Kruskal-Wallis test with Bonferroni’s correction for multiple comparisons was used for between-cluster comparison at baseline. The Wilcoxon Signed-Rank test and Friedman test were used for within-cluster comparison between 2-months and 4-months from baseline. Generalized Estimating Equations (GEE) adjusted for baseline values, age, and gender were used to analyze the intervention effect on intrinsic capacity, as well as functional ability during the 4-month study period. Analyses were performed using IBM SPSS Statistics for Windows, version 24 (IBM SPSS Statistics, Armonk, NY, USA: IBM Corporation).

## 3. Results

Diet quality is shown in Table 1 and Figure 2 as T-HEI throughout the study period for meals provided in retirement homes only (T-HEI_Diet_) and the addition of dietary supplements (T-HEI_Diet+PDS_). Detailed dietary intake throughout the study was also provided in Appendix B, Table A2, Table A3, Table A4 and Table A5. Participants were grouped into three different clusters based on dietary adherence similarity across study period. The overall mean T-HEI_Diet+PDS_ score across study were 75.7 ± 0.5, 64.5 ± 0.4, and 55.4 ± 0.7 for Cluster 1, Cluster 2, and Cluster 3, respectively. The Kruskal-Wallis tests showed that, diet quality among clusters were significantly different in the first [T-HEI_Diet_: *H* (2) = 30.635, *p* < 0.001; T-HEI_Diet+PDS_: *H* (2) = 34.678, *p* < 0.001], second [T-HEI_Diet_: *H* (2) = 40.371, *p* < 0.001; T-HEI_Diet+PDS_: *H* (2) = 45.173, *p* < 0.001], third [T-HEI_Diet_: *H* (2) = 39.509, *p* < 0.001; T-HEI_Diet+PDS_: *H* (2) = 42.610, *p* < 0.001], and fourth [T-HEI_Diet_: *H* (2) = 42.353, *p* < 0.001; T-HEI_Diet+PDS_: *H* (2) = 47.680, *p* < 0.001] month of the study. The addition of PDS significantly improved diet quality among participants in Cluster 1 in the second (Z = −3.184, *p* = 0.001), third (Z = −3.119, *p* = 0.002), and fourth (Z = −3.200, *p* = 0.001) month but not the first (Z = −0.80, *p* = 0.420) month of the study, compared to diet alone. Diet quality among participants in Cluster 2 in the first (Z = −2.953, *p* = 0.003), second (Z = −4.513, *p* < 0.001), third (Z = −3.323, *p* = 0.001), and fourth (Z = −4.784, *p* < 0.001) month of the study was also significantly improved with the addition of PDS. Participants in Cluster 3, who adhered less to the dietary intervention, by contrast, had T-HEI_Diet + PDS_ scores indifferent from T-HEI_Diet_ score until changes were observed in the third (Z = −3.467, *p* = 0.001) and the fourth (Z = −2.207, *p* = 0.027) month of the study (Figure 2).

Table 2 shows the baseline characteristics of the full samples and the comparison between clusters. The mean age of the total participants was 80.7 ± 1.1 years with the majority of them being female (44 cases, 74.6%). It was revealed that clusters were significantly different in age [*H* (2) = 8.352, *p* = 0.015], as well as CST [*H* (2) = 6.242, *p* = 0.044], GS [*H* (2) = 7.761, *p* = 0.021], and HGS performance [*H* (2) = 6.702, *p* = 0.035]. Cluster 3 was significantly older (*z* = −18.310, *p_adjusted_* = 0.013), lower in CST (*z* = 14.567, *p_adjusted_* = 0.050) and GS (*z* = 15.839, *p_adjusted_* = 0.041) performance than cluster 1. Cluster 2 was significantly higher in GS (*z* = 12.865, *p_adjusted_* = 0.047) and HGS (*z* = 13.715, *p_adjusted_* = 0.029) performance than cluster 2.

Figure 3, Figure 4, Figure 5 and Figure 6 show the within-cluster comparison of intrinsic capacity at two-month and four-month study periods. A significant trend of improvement from baseline was seen in MNA among participants in Cluster 1 [χ^2^ (2) = 7244, *p* = 0.027]. Significant improvement from baseline was observed in TMS at four-month of intervention (Cluster 2: Z = −2.613, *p* = 0.009) and CST at two-month (Cluster 1: Z = −2.267, *p* = 0.023) of intervention among participants who were higher adhered to dietary intervention; a significant decreased in HGS in four-month of intervention was observed in Cluster 1 (Z = −2.613, *p* = 0.009). Compared to the baseline, the CSR test significantly improved regardless of dietary intervention adherence.

The dietary intervention adherence effect on intrinsic capacity is shown in Table 3. After adjusting for covariates, significant interaction effects between higher dietary intervention compliance and time were observed in TMS (*p_interaction_* = 0.009) and MMSE (*p_interaction_* = 0.022). Significant cluster effects were observed in GS (*p_clusters_* = 0.034) and TMS (*p_clusters_* = 0.010). Significant time effects were observed in CST (*p_time_* < 0.001), TMS (*p_time_* = 0.031), and MMSE (*p_time_* = 0.022). After four months of intervention, ADL significantly improved among participants who were highly adhered to dietary intervention (*p_interaction_* = 0.034; Table 4).

## 4. Discussion

Healthy aging has been reframed as a holistic approach to developing and maintaining functional ability in older age rather than the aim for the absence of disease [6]. Intrinsic capacity, the composite of all the physical and mental reserves that an individual can draw on throughout his/her life course, is proposed to be the core of healthy aging. Studies monitoring the trajectories of intrinsic capacity in older adults suggest that intrinsic capacity declines progressively with age. It is also found that inferior intrinsic capacity predicts adverse health outcomes such as an increase in the risk of ADL disability, nursing home stay, and mortality [13,31]. Promoting capacity-enhancing behavior, healthy eating, in particular, to slow down or reverse the decline in intrinsic capacity and maintain functional ability, is, therefore, a public health priority facing population aging. Our study shows that improving diet quality is beneficial in intrinsic capacity enhancement, especially in the areas of vitality and locomotion, thus promoting ability in the ADL over time among older adults living in retirement homes.

Along with advancing age, changes in body composition, characterized by a gradual decline in skeletal muscle mass coupled with body fat increases, not only result in decreased locomotor resilience but also seriously affect vitality, conceptualized as the underlying resilience that contributes to more overt manifestation capacity including locomotion, cognition, sensory capacity, and psychological capacity [8]. Decreases in metabolic rate and physical activity owing to muscle mass loss consequently cause a decline in energy requirements. Diminished food consumption and nutrient intake followed by depleted energy intake may further compromise nutritional status among older adults since there is little evidence suggesting that nutrient requirements, both macro- and micro-nutrients, are decreased as well [32]. Nutrition is also an integral part of immune and antioxidative defense systems in which decreasing functioning is the general hallmark of muscular and neurological aging [14,33,34,35,36]. Vitamins and minerals are an integrated part of both enzymatic and non-enzymatic mechanisms in battling oxidative stress, while the antioxidative properties of omega-3 fatty acids and phytonutrients function in reducing the inflammatory response [37,38,39,40].

Malnutrition is considered one of the major culprits for decreased vitality in older age, while better diet quality has been suggested to be associated with a lower risk of malnutrition [7,41]. Dietary patterns of high quality, such as the Dietary Approach to Stop Hypertension (DASH) and Mediterranean Diet (MedDiet), have also been widely reported to potentially decrease the risk of frailty, which is a geriatric syndrome characterized by accumulation of deficits, conceivably the opposite of vitality [42,43,44,45]. Our previous study also found that older adults with better diet quality credited to dietary behavior compliant to DFGs and DGs of Taiwan assessed by T-HEI were associated with a lower prevalence of frailty [28]. Accumulating evidence also suggests the neuroprotective effects of healthy diets characterized by nutrient-dense food choices such as whole grains, dark or orange vegetables, fruits, as well as nuts and seeds that are rich in vitamins, minerals, phytonutrients, and omega-3 fatty acids against cognitive decline and depressive symptoms [46,47]. Their antioxidative and anti-inflammatory properties have been associated with better musculoskeletal functions and greater functional ability among older adults [44,45,48,49]. Older adults supplemented with PDS have also been reported to enhance antioxidative ability and innate immunity in subproject 3 [23,50].

According to the 2013–2016 Nutritional and Health Survey in Taiwan (NAHSIT), intake of nutrient-dense foods such as whole grains, fruits, as well as nuts and seeds was below DFGs and DGs suggestion among older Taiwanese, which might be associated with edentulism and decreased oral health [51,52,53,54]. The T-HEI score at baseline in our study was low, which evidently resulted from the omission of dark-and-orange color vegetables, fruits, as well as nuts and seeds, both deficits considered the key to undesirable diet quality. In this study, we intended to remove the barrier to healthy eating by providing texture-modified PDS in order to close the gap between suggestion and actual intake which then resulted in a significant increase in the diet quality. A “virtuous cycle” was observed as participants in Cluster 1 were higher in T-HEI_Diet_ at baseline, which indicated existing healthy dietary behavior among participants with higher dietary intervention compliance thus, a greater intervention effect was achieved. However, a decrease in intervention effect was observed during the winter, the fourth month of the study, which might be attributed to the lower dietary intervention compliance with frozen-serve PDS.

Performance in physical fitness is crucial to intrinsic capacity and functional ability. Cardiorespiratory endurance is a fundamental component of physical fitness. Muscle strength of both upper and lower extremities is important for numerous tasks in daily living, including lifting groceries and climbing stairs. Gait speed assessed agility and dynamic balance which represents the ability in performing maneuvering-required tasks such as getting up from a chair or getting off a bus. Postural steadiness and static balance, meanwhile, are important indicators for neurological and musculoskeletal status monitoring and fall risk management. Flexibility in the upper and lower extremities is central in performing the activity of daily living such as bathing, grooming, and dressing [55,56].

Within-cluster comparison results suggest that increased diet quality contributed to the enhancement of cardiorespiratory endurance and nutritional status among participants with higher dietary intervention compliance. A tendency toward improvement in HGS was observed in the highest dietary intervention compliance cluster at two-month of intervention but decreased at the four-month of intervention. After adjusting for baseline values and covariates, only TMS remained significantly improved among participants with improved diet quality over time. Our result aligns with previous work done by Chan and colleagues which suggested better diet quality was independently associated with higher seven-year cardiorespiratory fitness [57]. Besides, a greater significant improvement in lower extremities flexibility was observed. Performance on CST among participants with the highest dietary intervention compliance was significantly improved at two-month of intervention but the improvement did not sustain through four-month of interventions. After controlling for covariates, a significant intervention effect was observed in GS. A decreased performance over time, on the other hand, was observed in CST across clusters. Emerging evidence indicates the promising synergetic effect of combined nutrition and exercise intervention in muscle function and locomotor capacity strengthening [58,59]. The absence of exercise training may be the explanation for the lack of interventional effect on locomotor capacity in our study. Nonetheless, it is worth mentioning that enhanced TMS performance may be considered an improvement in the locomotor capacity, as cardiorespiratory endurance is considered a potential indicator of locomotion due to its reliability in predicting cardiorespiratory function and functional capacity in mobility [60].

Limited to the small sample size and statistical power, our study was unable to detect a significant change in psychological capacity in the generalized estimating equation model. Notwithstanding, a decrease in GDS scores was observed in a. within-cluster comparison among participants with improved diet quality. A within-normal-range decrease of cognitive capacity was seen in younger age participants but relatively stable among older age participants which might suggest a ‘normal cognitive aging’, defined as decline not meeting criteria for cognitive impairment, in our participants. A short interventional period and decreased dietary intervention compliance during winter as mentioned before, on the other hand, is another possible argument for the failure in cognitive performance improvement.

Sensory capacity is important in maintaining independent living. Impairment is not only associated with great health disparity but is also involved in cognitive and functional decline among older adults [61,62]. In our study, however, samples who were vision impaired and were unable to communicate due to hearing problems were excluded. We therefore cannot explore the effect of dietary intervention on the sensory domain, as well as the effect on the intact concept of intrinsic capacity, which is the major limitation of our study.

The unavailability of nutritional biomarkers put a limit on the validation of dietary interventional effect on nutritional status. However, the assessment in our study was carried out by certified professional and para-professional personnel, which ensures the accuracy of the data collected. Moreover, as the original study was designed to answer the specific research question of interest rather than intrinsic capacity, we are therefore unable to cover all the indicators in every domain of intrinsic capacity. Nonetheless, the indicators used in this study have been validated as clinically useful indicators of intrinsic capacity [8,12].

## 5. Conclusions

In conclusion, nutrition is one of the cornerstones of healthy aging. Adherence to healthy eating verified by T-HEI enhances vitality, the underlying resilience contributes to intrinsic capacity, and locomotion, thus promoting functional ability among older adults. Future studies focusing on the long-term effects of dietary nutrition on intrinsic capacity are appreciated.

## Figures and Tables

**Figure 1 nutrients-14-02225-f001:**
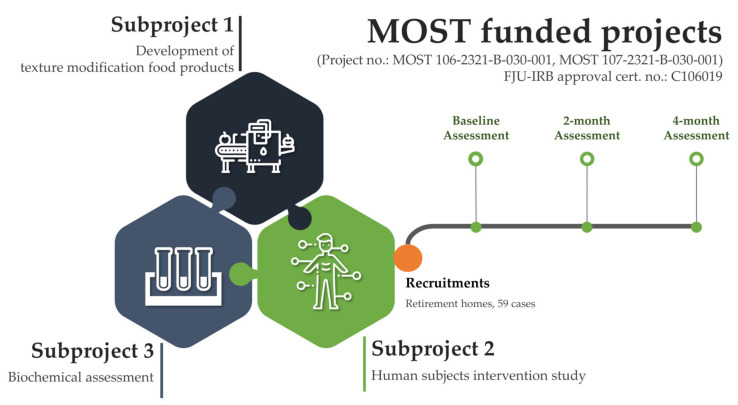
Data acquisition and study procedure.

**Figure 2 nutrients-14-02225-f002:**
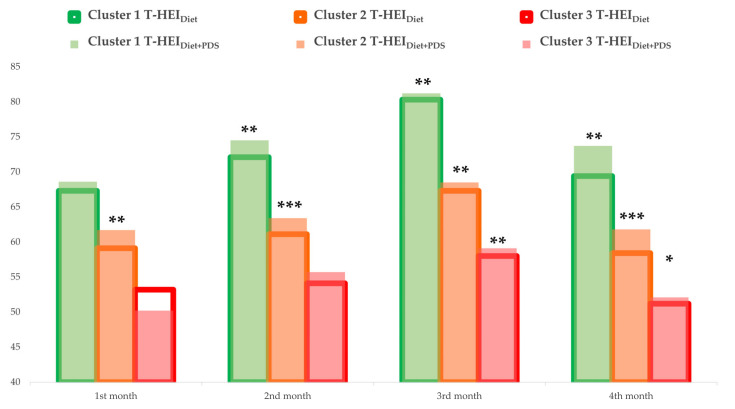
Bar chart for T-HEI mean changes during the intervention period. *, **, *** Indicated significant different from diet alone, compared by Wilcoxon Signed Rank test; **p* <0.05, ***p* <0.01, ****p* <0.001. T-HEI, Taiwanese Healthy Eating Index.

**Figure 3 nutrients-14-02225-f003:**
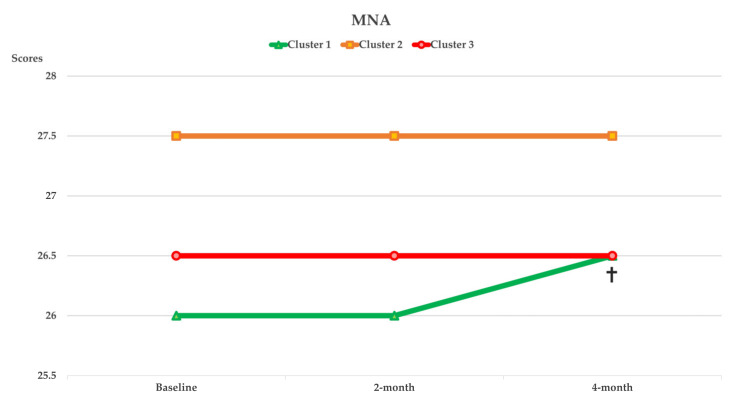
Curve chart for the within cluster comparison of median of intrinsic capacity in vitality captured by MNA, HGS, and TMS at two-month and four-month of intervention. *****, ****** Indicated significant different from baseline, compared by Wilcoxon Signed Rank test; ******p* < 0.05, *******p* < 0.01. † Indicated significant different trend from baseline, compared by Friedman test. HGS, handgrip strength; MNA, Mini Nutritional Assessment; TMS, two minutes steps.

**Figure 4 nutrients-14-02225-f004:**
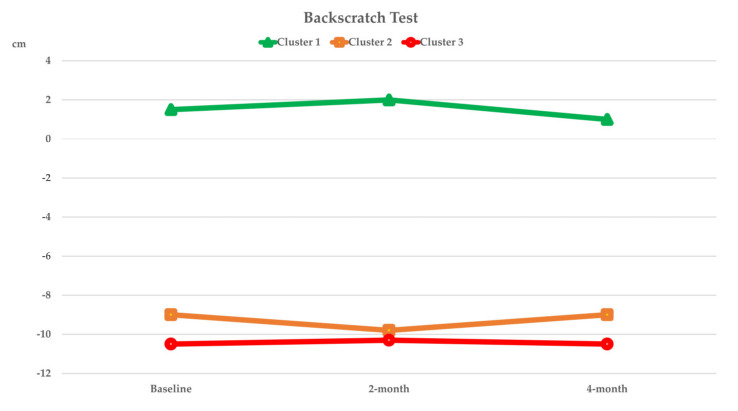
Curve chart for the within cluster comparison of median of intrinsic capacity in locomotion captured by BST, CSR, and CST at two-month and four-month of intervention. *****, ******, ******* Indicated significant different from baseline, compared by Wilcoxon Signed Rank test; ******p* < 0.05, *******p* < 0.01, ********p* < 0.001. BST, backscratch test; CSR, chair-sit-and-reach test; CST, Chair stand test.

**Figure 5 nutrients-14-02225-f005:**
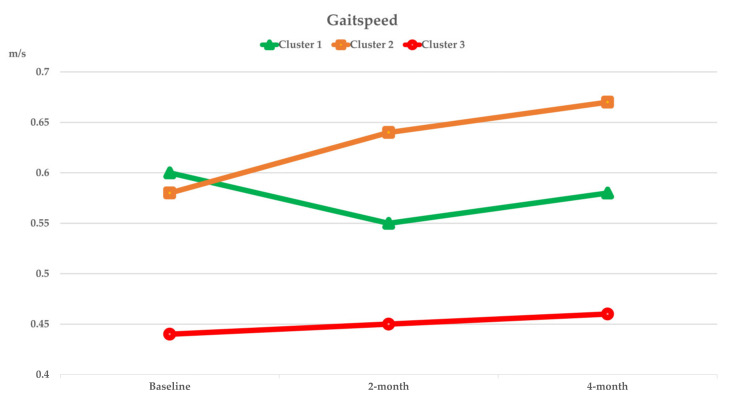
Curve chart for the within cluster comparison of median of intrinsic capacity in locomotion captured by GS and OFS at two-month and four-month of intervention. GS, gaitspeed; OFS, One foot standing test.

**Figure 6 nutrients-14-02225-f006:**
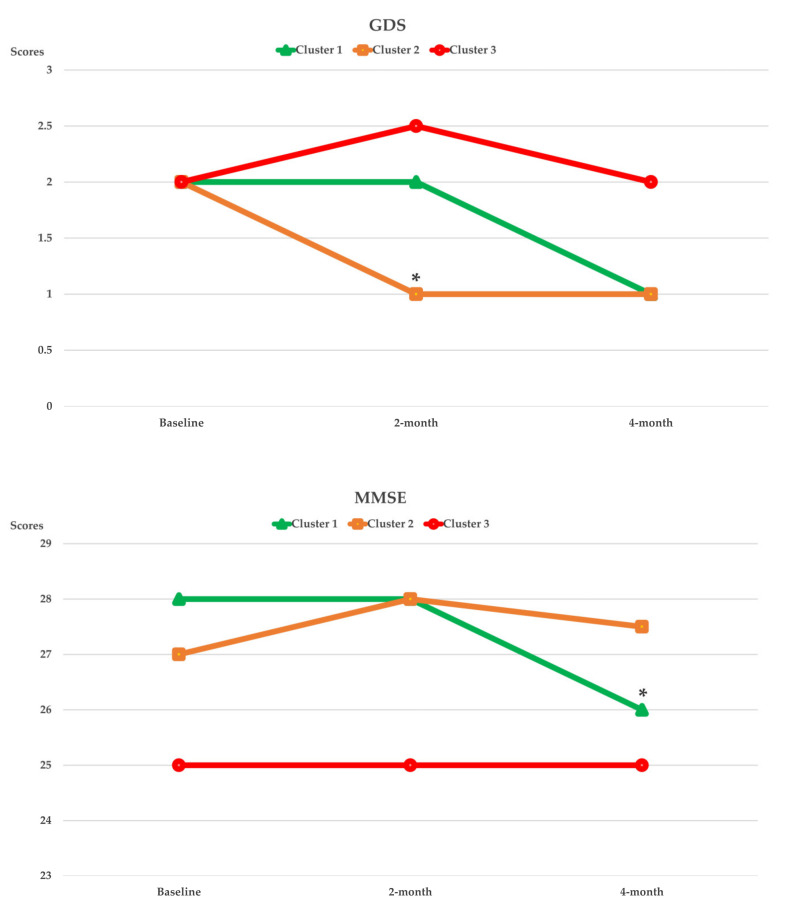
Curve chart for the within cluster comparison of median of intrinsic capacity in cognition and psychological capacity captured by GDS and MMSE at two-month and four-month of intervention. ***** Indicated significant different from baseline, compared by Wilcoxon Signed Rank test; ******p* < 0.05. GDS, 15-item Geriatric Depression Scale; MMSE, Mini Mental State Examination.

**Table 1 nutrients-14-02225-t001:** Diet quality comparison across dietary adherence clusters throughout the study, based on T-HEI score ^1^.

		All	Cluster 1 (*n* = 13)	Cluster 2 (*n* = 30)	Cluster 3 (*n* = 16)	*p*-Value ^2^
1st month	Diet	59.3 ± 0.9	67.3 ± 0.8 ^a^	59.1 ± 0.9 ^b^	53.2 ± 0.9 ^c^	<0.001
	Diet + PDS	60.1 ± 1.1	68.6 ± 1.2 ^a^	61.7 ± 1.0 ^b,^**	50.2 ± 1.4 ^c^	<0.001
2nd month	Diet	61.6 ± 1.0	72.1 ± 1.0 ^a^	61.1 ± 0.9 ^b^	54.1 ± 0.8 ^c^	<0.001
	Diet + PDS	63.8 ± 1.0 ***	74.5 ± 0.9 ^a,^**	63.4 ± 0.7 ^b,^***	55.7 ± 0.9 ^c^	<0.001
3rd month	Diet	67.6 ± 1.2	80.3 ± 0.3 ^a^	67.3 ± 1.1 ^b^	58 ± 1.5 ^c^	<0.001
	Diet + PDS	68.8 ± 1.2 ***	81.2 ± 0.3 ^a,^**	68.5 ± 1.0 ^b,^**	59.1 ± 1.2 ^c,^**	<0.001
4th month	Diet	58.9 ± 1.0	69.4 ± 0.4 ^a^	58.4 ± 0.9 ^b^	51.2 ± 0.5 ^c^	<0.001
	Diet + PDS	61.8 ± 1.1 ***	73.7 ± 0.5 ^a,^**	61.8 ± 0.9 ^b,^***	52.1 ± 0.5 ^c^^,^*	<0.001

Abbreviations: PDS, plant-based dietary supplements; T-HEI, Taiwanese Healthy Eating Index. ^1^ Values expressed as mean ± standard error of mean. ^2^ Based on Kruskal Wallis test with Bonferroni’s correction; ^a, b, c^ Nonidentical superscript indicated significant difference. *, **, *** Indicated significant different from diet alone, compared by Wilcoxon Signed Rank test; **p* <0.05, ***p* <0.01, ****p* <0.001.

**Table 2 nutrients-14-02225-t002:** Baseline characteristics of all participants and across baseline dietary adherence clusters ^1^.

		ALL (*n* = 59)	Cluster 1 (*n* = 13)	Cluster 2 (*n* = 30)	Cluster 3 (*n* = 16)	*p*-Value ^2^
Age		80.7 ± 1.1	76.2 ± 2.6 ^a^	80.3 ± 1.3 ^a,b^	85.1 ± 2.2 ^b^	0.015
BMI		24.7 ± 0.5	25.4 ± 1.5	24.1 ± 0.6	25.2 ± 0.8	0.481
T-HEI_Diet_ (0–100)		59.3 ± 0.9	67.3 ± 0.8 ^a^	59.1 ± 0.9 ^b^	53.2 ± 0.9 ^c^	<0.001
ADL (0–100)		95.2 ± 1.1	91.5 ± 3.9	97.2 ± 1.1	94.4 ± 1.8	0.216
IADL (0–24)		18.2 ± 0.7	19.5 ± 1.4	18.9 ± 1.0	15.9 ± 1.5	0.079
Vitality	MNA (0–30)	26.5 ± 0.3	25.5 ± 0.6	26.9 ± 0.4	26.4 ± 0.5	0.080
	HGS (kg)	20.9 ± 0.9	20.8 ± 1.6 ^a b^	22.7 ± 1.2 ^a^	17.7 ± 1.8 ^b^	0.035
	TMS (times)	73.5 ± 3.9	75.5 ± 11.9	76.3 ± 5.0	66.8 ± 5.8	0.582
Locomotion	BST (cm)	−9.6 ± 2	−4.0 ± 3.1	−12.6 ± 3.2	−8.4 ± 3.4	0.356
	CSR (cm)	−4.2 ± 1.6	−1.6 ± 3.3	−3.1 ± 2.2	−8.2 ± 2.9	0.192
	CST (times)	13.3 ± 0.8	14.3 ± 1.7 ^a^	14.2 ± 1.2 ^a,b^	10.6 ± 0.9 ^b^	0.044
	GS (m/s)	0.6 ± 0	0.7 ± 0.1 ^a^	0.6 ± 0 ^a^	0.5 ± 0.1 ^b^	0.021
	OFS (sec)	8.9 ± 1.3	12.7 ± 3.9	9.1 ± 1.7	5.4 ± 1.9	0.101
Psychology	GDS (0–15)	2.6 ± 0.3	2.7 ± 0.6	2.3 ± 0.4	3.0 ± 0.7	0.702
Cognition	MMSE (0–30)	25.7 ± 0.6	27.5 ± 0.7	25.6 ± 1.0	24.4 ± 1.0	0.074

Abbreviations: ADL, activity of daily living; BST, backscratch test; CSR, chair sit and reach; CST, chair stand test; GDS, 15-item Geriatric Depression Scale; GS, gaitspeed; HGS, handgrip strength; IADL, instrumental activity of daily living; MMSE, Mini-Mental State Examination; MNA, Mini Nutritional Assessment; OFS, one foot standing; T-HEI, Taiwanese Healthy Eating Index; TMS, two-minute steps. ^1^ Values expressed as mean ± standard error of mean. ^2^ Based on Kruskal Wallis test with Bonferroni’s correction; ^a, b, c^ Nonidentical superscript indicated significant difference.

**Table 3 nutrients-14-02225-t003:** Intervention effects on intrinsic capacity during the 4-month study period ^1,2^.

			Cluster 1 (*n* = 13)	Cluster 2 (*n* = 30)	Cluster 3 (*n* = 16)	*P* _clusters_	*P* _time_	*P* _interaction_ ^3^
Vitality	MNA	Baseline	25.5 ± 0.6	26.9 ± 0.4	26.4 ± 0.5	0.180	0.764	0.407
		2-month	25.9 ± 0.7	27 ± 0.4	26.5 ± 0.4			
		4-month	26.7 ± 0.6	26.6 ± 0.4	26.4 ± 0.5			
	HGS	Baseline	20.8 ± 1.6	22.7 ± 1.2	17.7 ± 1.8	0.704	0.284	0.126
		2-month	20.2 ± 1.5	22.1 ± 1.0	18.0 ± 1.7			
		4-month	18.9 ± 1.7	22.3 ± 1.1	18.2 ± 2.0			
	TMS	Baseline	75.5 ± 11.9	76.3 ± 5.0	66.8 ± 5.8	0.010	0.031	0.009
		2-month	75.0 ± 11.4	83.2 ± 4.4	66.4 ± 6.4			
		4-month	87.7 ± 9.1	88.8 ± 3.6	64.7 ± 6.2			
Locomotion	BST	Baseline	−4.2 ± 3.2	−12.6 ± 3.2	−8.4 ± 3.4	0.310	0.595	0.295
		2-month	−6.1 ± 4.2	−13.0 ± 2.9	−10.4 ± 4.2			
		4-month	−6.7 ± 4.2	−11.6 ± 2.9	−10.3 ± 4.1			
	CSR	Baseline	−1.6 ± 3.3	−3.1 ± 2.2	−8.2 ± 2.9	0.090	0.093	0.989
		2-month	14.1 ± 2.2	9.7 ± 1.8	−0.6 ± 3.9			
		4-month	12.4 ± 3.2	7.8 ± 2.0	−2.2 ± 3.7			
	CST	Baseline	14.3 ± 1.7	14.2 ± 1.2	10.6 ± 0.9	0.878	<0.001	0.079
		2-month	17.3 ± 2.1	15.3 ± 0.8	12.0 ± 1.2			
		4-month	14.8 ± 2.0	13.9 ± 1.0	11.4 ± 1.1			
	GS	Baseline	0.7 ± 0.1	0.6 ± 0	0.5 ± 0.1	0.034	0.110	0.105
		2-month	0.6 ± 0.1	0.6 ± 0	0.5 ± 0			
		4-month	0.6 ± 0.1	0.6 ± 0	0.4 ± 0			
	OFS	Baseline	12.7 ± 3.9	9.1 ± 1.7	5.4 ± 1.9	0.139	0.162	0.602
		2-month	10.9 ± 3.5	10.9 ± 1.9	4.9 ± 2.0			
		4-month	13.6 ± 3.6	11.5 ± 1.9	5.3 ± 1.9			
Psychology	GDS	Baseline	2.7 ± 0.6	2.3 ± 0.4	3.0 ± 0.7	0.268	0.545	0.779
		2-month	2.0 ± 0.4	1.6 ± 0.3	2.6 ± 0.4			
		4-month	1.9 ± 0.5	1.8 ± 0.3	2.9 ± 0.6			
Cognition	MMSE	Baseline	27.5 ± 0.7	25.6 ± 1.0	24.4 ± 1.0	0.078	0.028	0.022
		2-month	26.7 ± 0.9	25.4 ± 1.0	24.1 ± 1.0			
		4-month	25.7 ± 1.1	25.5 ± 1.0	23.9 ± 0.9			

Abbreviations: BST, backscratch test; CSR, chair sit and reach; CST, chair stand test; GDS, 15-item Geriatric Depression Scale; GS, gait speed; HGS, handgrip strength; MMSE, Mini Mental State Examination; MNA, Mini Nutritional Assessment; OFS, one foot standing; T-HEI, Taiwanese Healthy Eating Index; TMS, two-minute steps. ^1^ Adjusted for age, baseline intrinsic capacity value, gender, and baseline T-HEI. ^2^ Values expressed as mean ± standard error of mean. ^3^ Clusters × time effects.

**Table 4 nutrients-14-02225-t004:** Intervention effects on functional ability during the 4-month study period ^1,2^.

		Cluster 1 (*n* = 13)	Cluster 2 (*n* = 30)	Cluster 3 (*n* = 16)	*P* _clusters_	*P* _time_	*P* _interaction_ ^3^
ADL	Baseline	91.5 ± 3.9	97.2 ± 1.1	94.4 ± 1.8	0.471	0.498	0.034
	2-month	90.4 ± 4.2	97.5 ± 1.1	94.4 ± 1.8			
	4-month	96.2 ± 1.7	95.2 ± 1.6	93.1 ± 2			
IADL	Baseline	19.5 ± 1.4	18.9 ± 1	15.9 ± 1.5	0.792	0.177	0.139
	2-month	17.8 ± 1.8	18.7 ± 1.1	15.9 ± 1.5			
	4-month	18.8 ± 1.5	18.9 ± 1.1	15.8 ± 1.5			

Abbreviations: ADL, ADL, activity of daily living; IADL, instrumental activity of daily living; T-HEI, Taiwanese Healthy Eating Index. ^1^ Adjusted for age, baseline intrinsic capacity value, gender, and baseline T-HEI. ^2^ Values expressed as mean ± standard error of mean. ^3^ Clusters × time effects.

## Data Availability

The data presented in this study are available upon request pending application to and approval by research team of “New Strategies for Developing Healthy, Aging and Happy Diet”. The data are not publicly available due to ethical and legal consideration under Human Subjects Research Act of Taiwan.

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
