# Peer review of "Healthy Eating Enhances Intrinsic Capacity, Thus Promoting Functional Ability of Retirement Home Residents in Northern Taiwan"

_nutrients, 2022, doi:10.3390/nu14112225_

Round 1

Reviewer 1 Report

Dear Authors

this is a very important topic relating to healthy aging and determining dietary quality improvement. However, I have a question about the description of the study sample, it was stated that they are people from retirement homes. What age are they? What is their health status? Have there been any diseases found in the people in the study? To what extent could these factors be crucial to the results. Also, was ethics committee approval obtained for this type of study.

Author Response

Dear Reviewer 1:

Thank you for your suggestion. Attached is our reply to your suggestions. 

Best regards,

Reviewer 2 Report

The manuscript presents an interesting study based on the intervention effects of improved dietary quality on intrinsic capacity. 

During the reading of this article, I had minor suggestions:

Add to the introduction section:

  1. How much do people live on average now?
  2. In which part of the world do people live the longest and the shortest, and why? Figure 2 has poor visibility.

Figure 2 has poor visibility.

Author Response

Dear Reviewer 2:

Thank you very much for your suggestions. Attached is our reply to your suggestions. Best regards,

Reviewer 3 Report

The study requires nutritional biomarkers, since in the current state it is very descriptive and demonstrates a positive association between the parameters studied and the nutritional intervention. However, this association does not allow describing a quantitative trend or showing an effect from the point of view of a biomarker measured in older adults. This suggest that the journal is not appropriate for this study. Some minor observations:

-Please include in materials and methods a figure or diagram that shows the sequence of the study.
-Please include a bibliographic citation of line 90-93.
-Improve the quality of the figure2.

Author Response

Dear Reviewer 3:

Thank you very much for your suggestions. Attached is our reply to your suggestions. 

Best regards,

Round 2

Reviewer 3 Report

Accept

Author Response

Dear Reviewer:

Attached is our point-by-point response to the comments for our manuscript. Thank you so much for the valuable suggestions which allow us to improve our work to a better shape. 
